# Factors Influencing the Concentration of Exhaled Nitric Oxide (FeNO) in School Children Aged 8–9-Years-Old in Krakow, with High FeNO Values ≥ 20 ppb

**DOI:** 10.3390/medicina58020146

**Published:** 2022-01-18

**Authors:** Marta Czubaj-Kowal, Grzegorz Józef Nowicki, Ryszard Kurzawa, Maciej Polak, Barbara Ślusarska

**Affiliations:** 1Department of Paediatrics, Stefan Zeromski Specialist Hospital in Krakow, Na Skarpie 66 Str., 31-913 Krakow, Poland; 2Department of Pediatrics, Faculty of Medicine and Health Sciences, Andrzej Frycz Modrzewski Krakow University, Gustawa Herlinga-Grudzińskiego 1 Str., 30-705 Krakow, Poland; 3Department of Family and Geriatric Nursing, Medical University of Lublin, Staszica 6 Str., 20-081 Lublin, Poland; gnowicki84@gmail.com (G.J.N.); basiaslusarska@gmail.com (B.Ś.); 4Department of Alergology and Pneumonology, Institute of Tuberculosis and Lung Disorders, Prof. Jana Rudnika 3B Str., 34-700 Rabka-Zdrój, Poland; ryszard.kurzawa@gmail.com; 5Department of Epidemiology and Population Studies, Jagiellonian University Medical College, Grzegórzecka 20 Str., 31-531 Krakow, Poland; maciej.1.polak@uj.edu.pl

**Keywords:** fractional exhaled nitric oxide (FeNO), children, airway disease, treatment, body mass index, tobacco smoke exposure, prediction

## Abstract

*Background and Objectives*: Measurement of fractional exhaled nitric oxide (FeNO) concentration is currently used as a non-invasive biomarker to assess airway inflammation. Many factors can influence the FeNO level. However, there have been no reports concerning factors attributed to FeNO levels in different age groups of children, especially those with high FeNO values. Therefore, this study aimed to assess the influence of selected factors on nitric oxide concentration in exhaled air in children aged 8–9 attending class 3 of public primary schools in Krakow with high FeNO values ≥ 20 ppb. *Materials and Methods*: The population-based study covered all third-grade pupils attending primary schools in the city of Krakow. Five thousand, four hundred and sixty children participated in the first screening stage, conducted from October 2017 to January 2018. Then, 792 participants with an FeNO level ≥ 20 ppb were selected. Finally, those selected pupils were invited to participate in the second stage, diagnostic, in April 2018. Four hundred and fifty-four children completed the diagnostic stage of the study, and their data was included in the presented analysis. *Results and Conclusions:* Significantly higher FeNO levels were observed in children diagnosed with the following diseases: asthma, allergic rhinitis, atopic dermatitis, and allergy (*p* < 0.05). In addition, it was observed that a higher FeNO concentration characterised children taking antihistamines compared to children not taking those medications (*p* = 0.008). In multivariate models, we observed that regardless of sex, age, BMI value, home smoking, and whether they were taking medications, children who had allergic rhinitis, or atopic dermatitis, or allergies had significantly higher FeNO levels. The strongest relationship was noted with allergic diseases. The results of our study may be of importance to clinicians when interpreting FeNO results, for example, when making a therapeutic decision.

## 1. Introduction

Nitric oxide (NO) is endogenously released in the respiratory tract through the oxidation of the amino acid L-arginine by three nitric oxide synthases (NOS), which is expressed in numerous cell types such as epithelial cells, inflammatory cells (macrophages, neutrophils and mast cells), nerve airways, and vascular endothelial cells [1]. There are three different NOS isoforms: two constitutive isoforms, neuronal NOS (nNOS) and endothelial NOS (eNOS), and one inducible NOS (iNOS). INOS expression is enhanced by inflammatory stimulation [2]. The production of NO depends on many local factors, including genetic and epigenetic variants, the amount and activity of enzymes responsible for NO production, the concentration of substrates, the level of oxidative stress, and the rate of its uptake by antioxidant molecules. In the lungs, low levels of NO from nNOS and eNOS mediate a variety of physiological responses, including lung development, airway smooth muscle relaxation, bronchial protection, and ciliary mobility, while high levels of NO produced by iNOS appear to be involved in non-specific host defence mechanisms and chronic inflammation [1,2,3]. Fractional exhaled nitric oxide (FeNO) is currently used as a non-invasive biomarker to assess airway inflammation in various diseases [2]. NO is measured non-invasively, the test itself causes minimal discomfort to the patient, results are available within minutes of taking the measurement, while devices and protocols for FeNO measurement have been developed and refined since 1991 [2,4].

Most importantly, measuring FeNO can be performed independently by the patient or their caregiver. As the research results by Lo et al. have proven [5], a short training contributed to obtaining as many as 77% of well-performed FeNO measurements by the child or their caregiver in a group of 612 children. In addition, research results have confirmed the correlation between FeNO levels and: eosinophilic airway inflammation (blood eosinophil count, sputum count), serum eosinophil cationic protein, IgG concentration, and the number of positive skin allergy point tests. Moreover, FeNO concentration is associated with the response to oral and inhaled corticosteroids, bronchial hyper-reactivity, lung function, and asthma symptoms in children and adults [6,7].

Many factors may influence the level of FeNO [8,9]. The confounding factors may be patient-related, such as genetics, sex, weight and height, diet, medication (e.g., anti-inflammatory drugs), active and passive smoking, and atopy [10,11]. Contact with allergens is associated with higher FeNO levels, although on the other hand, FeNO may be lower in the early phase of the allergic response [12]. FeNO levels may decline during bronchospasm and exposure to tobacco smoke [13,14]. However, they may increase in certain conditions such as allergic rhinitis (AR), eosinophilic bronchitis (EB), atopic dermatitis, and adolescents with asthma [15], or the case of airway hyper-responsiveness without asthma [16]. Studies have shown that both active and passive smoking reduces the level of FeNO in healthy adults and patients with asthma and children with asthma, regardless of the presence of allergies [14]. Another important factor influencing the FeNO value is physical activity. Research results suggest that intense physical activity correlates with higher FeNO values in less active adolescents and children [17,18]. The level of FeNO may also be influenced by viral infections of the respiratory tract [19]. Age is an important factor influencing the FeNO level, especially in children [20], although the correlation between age and sex and FeNO level is not fully understood [21,22,23]. During childhood, FeNO physiologically increases with age and growth. It has been shown that the FeNO value increases linearly between the ages of 6 and 14 in girls, and between 6 and 16 years of age in boys, to reach a plateau in adulthood [24,25]. Race and ethnicity also appear to influence FeNO levels [25,26,27]. Healthy Hispanic children have higher FeNO levels than white non-Hispanic children [28], while healthy black children have higher FeNO levels than healthy Hispanic and white non-Hispanic children [27]. A recent study found that African American children with the most severe form of asthma have higher mean FeNO values at baseline, during disease flare, and after a short course of oral corticosteroids than non-African American (i.e., white and Hispanic) children, despite no differences in disease severity, lung function, and doses of treatment with inhaled corticosteroids [29]. Another large group influencing the FeNO value is the measurement conditions, or the analyser used [30].

Research on the usefulness of FeNO in diagnosing and monitoring the course of respiratory diseases has been ongoing for almost 30 years. However, the results still raise controversy about the usefulness of this marker, especially in children, mainly due to the factors influencing its concentration and the heterogeneity of the available studies. There are still no reports on the influence of various factors on the level of FeNO in children with high FeNO values. Therefore, this study aimed to assess the influence of selected factors on nitric oxide concentration in exhaled air in children aged 8–9 attending class 3 of public primary schools in Krakow with high FeNO values ≥ 20 ppb.

## 2. Materials and Methods

### 2.1. Study Design and Participants

This population-based study covered all third-grade students, that is, children aged 8–9, attending primary schools in the city of Krakow (*n* = 199) in the 2017–2018 school year. The study consisted of two stages (Figure 1). Five thousand, four hundred and sixty children participated in the first screening stage, conducted from October 2017 to January 2018. Seven hundred and ninety-two participants with an FeNO level of ≥20 ppb were selected amongst the 5460. The selected group was invited to participate in the second stage—a diagnostic one, conducted between 9 and 25 April 2018. During this stage, participants’ parents/legal guardians completed a questionnaire. From the selected group of 792 children, 454 children completed the study at the diagnostic stage, which accounted for 57.32% of children enrolled in the clinical stage and 5.86% of all children attending third grade in the city of Krakow. Three hundred and thirty-eight children who qualified for the screening stage did not participate in the diagnostic stage as: they did not attend the diagnostic stage (*n* = 295), or they changed their place of residence (*n* = 43). Therefore, the analyses included FeNO measurements obtained in the diagnostic stage.

The researchers collecting data consisted of specifically instructed physicians. As part of the screening stage, trained physicians met with the children in each class on two separate occasions. During the first meeting, the children were provided with a written explanation describing the purpose, method of the examination, and instructions on preparing the child for the examination. The parent’s/legal guardian’s consent for their children’s participation in the study was included as a separate sheet of paper. Then, the children were asked to provide this information to their parents/legal guardians. The information package for parents/legal guardians contained the following information: e-mail addresses and telephone numbers of the people who had designed the study for parents/legal guardians to contact and clarify any doubts. During the second meeting, informed consent signed by the parents/legal guardians was collected from the children, and an examination performed on those children who provided this signed consent. The research as part of the screening stage was conducted in the school nurse’s office. For the second diagnostic stage, individual invitations were sent to children whose FeNO in the diagnostic stage was ≥20 ppb.

The second stage of the study took place in the Outpatient Clinic of the Paediatric Ward, Stefan Żeromski Specialist Hospital in Krakow. Prior to the study, the parents/legal guardians were asked to provide their informed consent to participate in the second stage of the study. After providing this consent, the parents were asked to fill in the questionnaire. The research included two groups of children: healthy pupils and those with diagnosed respiratory diseases. The presented data constituted a part of the research project led by the Department of Paediatrics, Stefan Żeromski Specialist Hospital in Krakow, Poland, as a part of the campaign to improve air quality “Let us be together in the fight for clean air in Krakow” piloted by the Municipality of Krakow. The Bioethics Committee Institute of Tuberculosis and Lung Diseases (KB-26/2018 of 19 April 2018) provided the approval for the study.

### 2.2. Measurement of FeNO

The FeNO measurements were sampled, in accordance with the guidelines of the American Thoracic Society (ATS) [15,19,31] using the online method, with the Hyp’AirFeNO electrochemical analyser (MediSoft, Sorinnes, Belgium). The analyser used, assured repeatable measurements of FeNO in the range of 0–600 ppb without a need for any external calibration. The electrochemical breath analyser transforms gas concentration into electrical signals. Respondents exhaled via disposable mouthpieces at a continuous flow of 50 mL/s for 6 s. The apparatus was attuned and employed according to the manufacturer’s directions and applied in conjunction with a computer. The ExpAir software delivered an automatic recording of the data, automatic analysis of results, data logging, and a report. This device is easy to use, semi-portable, fast, and inexpensive, compared to other analysers based on the chemiluminescent and laser method. Its characteristics include good reliability and repeatability as well as a high level of collaboration with patients, especially paediatric ones. Moreover, the examination was painless and non-invasive for the participating children.

The participants were asked to abstain from eating, drinking, and physical activity, and avoid exposure to tobacco fumes. The examinations were conducted between 8:00 a.m. and 2:00 p.m. Prior to testing, the paediatric participants were tutored on how to perform the test, as well as checking their aptitude to implement the procedure. Each patient provided their FeNO measurement twice. The first attempt aimed at teaching the child how to perform the test, and the second constituted a diagnostic measurement; this outcome was taken for subsequent analysis. The FeNO result was outlined in ppb values (parts per billion, 10^−9^), which is a dimensionless description of the ratio of two values. The upper limit or normal value range of the FeNO in paediatric respondents was established at 20 ppb [15,19,32,33].

### 2.3. Anthropometric Measurements

Anthropometric measurements of height and weight were taken from all children. Bodyweight (in kg) was measured to an accuracy of 0.1 kg with a platform scale and height (in cm) also to one decimal place using an altimeter. Body mass index (BMI) was defined as body weight (kg) divided by height squared (m^2^): BMI = kg/m^2^. BMI categories were determined using cut-off points proposed by the International Obesity Task Force (IOTF): thinness (<17 kg/m^2^), normal weight (≥17 and <25 kg/m^2^), overweight (≥25 and <30 kg/m^2^), and obesity (≥30 kg/m^2^) [34,35,36].

### 2.4. Other Data

The remaining variables—age, sex, medical history, home exposure to tobacco smoke —were collected using standard questionnaires from the child’s parents during the diagnostic stage of the study. The parents/legal guardians of the children were asked to enter whether their child was diagnosed with the following diseases: asthma, allergic rhinitis (AR), atopic dermatitis, and allergy. The parents/legal guardians were also asked to enter the names of the child’s medications. The medications were then grouped into inhaled glucocorticosteroid (ICS), antihistamine, leukotriene receptor antagonist (LTRA), and intranasal corticosteroids.

### 2.5. Statistical Analysis

Data were expressed as mean and standard deviation (SD) or as median (interquartile range, IQR) as appropriate. The Shapiro–Wilk test was used to assess conformity with a normal distribution. The FeNO distribution was compared between two groups using the Mann–Whitney U test. The Spearman rank correlation was used to assess the relationship between FeNO and age. Multiple linear regression was used to assess the significant predictors of FeNO. The results analysis is presented as beta coefficient with 95% CI. The four separately multivariable models which were performed included asthma or allergic rhinitis or atopic dermatitis or allergy. Due to the right-skewed distribution of FeNO, logarithm transformation was applied. Two-sided *p*-values < 0.05 were considered statistically significant. Statistical analyses were conducted using IBM Corp. Released 2017. IBM SPSS Statistics for Windows, Version 25.0. (Armonk, NY: IBM Corp).

## 3. Results

### 3.1. Characteristic of Participants

Table 1 presents the characteristics of the study group. Four hundred and fifty-four children participated in the study. The mean age in the study group was 8.9 ± 0.46 years. All surveyed children lived within the administrative boundaries of the city of Krakow. The majority of the study group were boys (52.86%; *n* = 240). In the study group, the highest significant number of children suffered from allergy (49.5%; *n* = 225) and allergic rhinitis (31.72%; *n* = 144). In addition, the examined children most often took antihistamines (18.72%; *n* = 85).

### 3.2. Relationship between the Analysed Variables and FeNO Concentration

Table 2 presents the analysis of the relationship of selected variables with the FeNO concentration. Significantly higher FeNO levels were observed in children diagnosed with the following diseases: allergic rhinitis, atopic dermatitis, allergy, and asthma (*p* < 0.05). In addition, it was observed that children who took antihistamine medications also had a higher concentration of FeNO than those who did not take such medications (*p* = 0.008). The remaining variables did not differentiate the studied group in terms of FeNO concentration.

### 3.3. Relationship between the FeNO Level and Selected Variables in Multivariate Models

Table 3 presents multivariate models assessing the relationship between the concentration of FeNO (LogFeNO) and the investigated factors affecting its level. Regardless of sex, age, BMI, home smoking, or medications, significantly higher FeNO levels were observed in children suffering from allergic rhinitis, atopic dermatitis, or allergic diseases. The strongest relationship was with allergic diseases. However, no significant association was found between FeNO and the occurrence of asthma, sex, age, BMI categories, home smoking, and medications.

## 4. Discussion

Measurement of FeNO is a useful method of detecting the severity of airway inflammation [3]. The method of measuring FeNO has been standardised. In recent years this non-invasive test has been widely used in diagnosing asthma, the monitoring of airway inflammation, and the detection of asthma overlap in patients with chronic obstructive pulmonary disease (COPD). Research is also being conducted into the clinical application of FeNO measurements in other respiratory diseases such as COPD and interstitial lung diseases. Currently, there is some confusion about the importance of measurement and interpretation of results in clinical practice. The results of FeNO measurement, apart from factors related to the measurement conditions, are influenced by individual and environmental factors; however, none of the factors was considered a critical factor disturbing the interpretation of the measurement results [37]. Our research in a group of children with high FeNO values showed that diseases such as asthma, AR, atopic dermatitis, and allergy, as well as taking antihistamines, resulted in a higher FeNO score. However, the remaining analysed variables did not significantly differentiate the studied group. In multivariate models, we found that, regardless of sex, age, BMI, home smoking, or medications, children who had AR or atopic dermatitis or allergies had significantly higher FeNO levels. The results of our study may be of importance to clinicians when interpreting FeNO results, for example, when making a therapeutic decision.

As the lumen area of the airways increases with a child’s age in childhood, FeNO physiologically increases with a child’s age and growth [19]. A recently published study by Garcia et al. [28] conducted in a cohort of 1791 healthy school-aged children (mean age 8.4 ± 0.65 years at recruitment) demonstrated that the trajectory of FeNO change was the same in girls and boys until adolescence and early adolescence (~11.5 years). After reaching this period, FeNO levels continued to rise in boys, while in girls, they appeared to stabilize. The authors suggest that the change in FeNO trajectory may be related to hormones. Zhang et al. [38] in a longitudinal study of 3607 students in Southern California, observed a difference in the level of FeNO depending on age. Their study showed that children who were nine years old and younger had significantly lower FeNO levels than those over nine years old. The obtained difference coincides with early puberty and the action of sex hormones. In the authors’ research, no significant correlation was observed between age and FeNO level. However, we conducted our research in a group of children aged 8–9, so before puberty, therefore, since the studied group was homogeneous, the age of the respondents should not affect the conclusions we obtained.

Sex may affect FeNO levels, but research results are not consistent. In the studies of Linn et al. [39] (conducted in a group of 2568 healthy children aged 7–10), Zhang et al. [38] or Rachel et al. [40] (conducted in a group of 352 healthy children and children with respiratory diseases, aged 4–17), FeNO was independent of sex. On the other hand, Zhang et al. [41] in a study conducted with a group of 219 healthy children (mean age 10.16 ± 2.8 years) showed a relationship between sex and FeNO levels. However, no such relationship was observed in our research. Thus, the issue of changes in FeNO levels depending on sex requires further clarification.

Many factors impact upon the level of FeNO and pathological changes in respiratory diseases is one of them. Elevated FeNO levels occur in asthma, pneumonia, viral infections, AR, COPD, and decreased levels in cystic fibrosis, pulmonary hypertension, fixed cilia, and acute respiratory distress syndrome (ARDS) [42,43,44,45]. It was shown that patients with asthma, even those with mild disease, exhaled FeNO at higher concentrations, in correlation with high levels of NO synthesis induced expression in the respiratory epithelium [45]. The main basis of elevated FeNO levels is that NO modulates airway hyper-responsiveness [46] and is associated with eosinophilic inflammation [47]. Atopy, in which IgE antibodies specific for environmental antigens are detected, is an important confounding factor for FeNO values [15,19] while FeNO levels have been shown to be higher with more sensitising antigens [48,49]. Ma’Pol et al. [50] demonstrated a relationship between FeNO levels and allergy to house dust mites and cats, regardless of allergic symptoms. The pathogenic mechanism of allergic rhinitis is inflammation of the airways with eosinophilic infiltration caused by allergens as TH2 lymphocytes and cytokines, eosinophils and mast cells attack the nasal mucosa. Therefore, the release of NO synthesis from epithelial cells causes higher NO levels in the upper and lower respiratory tract [51]. Several research studies have disclosed elevated FeNO levels in adults and children with AR, although the published data on measuring FeNO in children with AR is not as broad [52,53]. The elevated FeNO levels in AR patients likely reflects an expansion of airway inflammation, a feature of the well-known “united airway” concept. It was found that the diffusing capacity of NO in the bronchial wall of symptomatic AR patients was greater compared to healthy controls. This fact may reveal variations in the physical properties of the bronchial mucosa induced by subclinical lower respiratory tract inflammation in AR [54]. A meta-analysis by Lu et al. [55] showed a mild elevation of FeNO in patients with COPD. This disease is mainly characterised by over-expression and infiltration of neutrophils, macrophages, and T cells [56]. Several inflammatory cytokines are then produced and released, and the process may involve the synthesis of iNOS by macrophages, epithelium, and airway smooth muscle [57]. Rachel et al. [40] conducted studies in a group of healthy children with cystic fibrosis, asthma, AR, and asthma and AR. They found elevated FeNO levels in patients with asthma, AR, and asthma and AR combined. The authors’ research showed that children with the analysed diseases had a significantly higher level of FeNO than healthy children.

Moreover, in multivariate models, we observed that children diagnosed with AR or atopic dermatitis, or allergies had significantly higher FeNO levels, regardless of sex, age, BMI, home smoking, or medications. In the multivariate models, however, no significant relationship was found between FeNO and the occurrence of asthma. Few studies analysing a clinically significant change in FeNO levels in respiratory diseases are available, and the results obtained in the published studies differ. As the authors’ results show, FeNO levels differ significantly in healthy patients and patients with respiratory diseases, so further studies on larger cohorts are needed to consider factors influencing FeNO levels.

The results of studies by other authors report that the FeNO level may be used to assess the effectiveness of anti-asthma treatment [58]. It has been shown that FeNO in patients previously untreated with steroids is a reliable predictor of ICS response [59]. In patients with higher baseline FeNO levels, its measurement was able to predict the response to ICS in both adults [60] and children [61]. Glucocorticosteroids can inhibit the impact of induced NO synthase, so after treatment there will be a decrease in FeNO level, which may significantly decrease after 2–3 days of ICS treatment, and the maximum effect can be achieved after 2–4 weeks, from which the level of FeNO may be predicted [20]. More controversial is the effect of LTRA on FeNO levels, as some studies have shown a rapid and sustained reduction inFeNO levels after treatment [62,63], while others have failed to find this association [64,65]. In vitro studies have shown that NO synthase activity can be reduced by antihistamine H1 therapy [66]. Animal studies have shown that histamine released by mast cells plays an important role in the production of FeNO and enhances bronchial hyper-responsiveness [67]. In vivo, levocetirizine has been shown to reduce FeNO after three months of treatment in children with mite allergy [68]. In our study, we observed that children treated with antihistamines have significantly higher FeNO levels than children not taking such drugs. This observation can be explained by the fact that we did not collect data on the duration of use of antihistamines, and the duration of use of these drugs may play a role in the FeNO value.

The studies conducted so far indicate that both active and passive smoking may be an important factor influencing the level of FeNO. Research results indicate that heavy smokers have a reduced FeNO level compared to non-smokers [48,69]. Kharitonov et al. [70] and Schilling et al. [71] noted a rapid decrease in FeNO levels after having a cigarette in their research. This is likely due to changes in endothelial function and pulmonary artery structure induced by tobacco smoke, such as inhibition of NOS enzymes, which may contribute to the development of respiratory and cardiovascular disease in smokers [72,73]. Merianos et al. [74] conducted a cross-sectional study of a large group of American teenagers. They found that adolescents with low serum cotinine levels had lower FeNO levels than adolescents with no or minimal cotinine levels. Multivariate analyses showed that tobacco smoke exposure in the environment for at least 2 h a day was negatively and significantly associated with FeNO levels, regardless of age and sex [75]. Laoudi et al. [76] showed that second-hand smoke reduces FeNO levels and may be the primary determinant of FeNO levels in untreated children with asthma. The results obtained in our study did not confirm the relationship between exposure to tobacco smoke and FeNO level. Like our observations, other authors also showed that there were no differences in FeNO levels in the group of children with asthma exposed to and not exposed to tobacco smoke [77,78]. Sundy et al. [79] observed only a weekly correlation between FeNO and serum cotinine levels in people with cotinine levels between 1 and 25 ng/mL—concentrations consistent with exposure to passive smoking. The obtained results can be explained by the fact that the researched group of children may have had lower exposure to tobacco smoke compared to active smokers. Moreover, parents and guardians of children were strongly prone to understating the risk concerning their children’s passive exposure to cigarette smoke. This opinion was supported by Boyaci et al. [80] and Howrylak et al. [81]. Therefore, in our group of children who were classified as children not exposed to passive smoking, there could be children who had contact with cigarette smoke, which could have influenced our results.

Obesity has been shown to be associated with impaired pulmonary function and FeNO in adults [82,83]. However, the current literature focusing on the effects of overweight and obesity on FeNO levels in children remains controversial. Research by Ai et al. [84] in a group of 481 children aged 6–15 years revealed that the level of FeNO was significantly positively correlated with the BMI index. Interesting results were published by Duong-Quy et al. [85]. They assessed the association between nasal FeNO and anthropometric features in children with AR and a healthy control group. Their research showed that in the group of healthy children, nasal FeNO was not significantly correlated with BMI, while in children with AR, this correlation was positive and statistically significant. On the other hand, Yao et al. [86] found that increased BMI was negatively associated with FeNO levels in children with atopy but not in children without atopy. The authors’ research did not show any relationship between the BMI value and the FeNO level. Discrepancies between studies may be due to variability in sample size, age, population diversity, ethnicity, and the tools used to assess overweight and obesity in children.

### Strengths and Limitations

The advantages and disadvantages of this study deserve consideration. First, an extensive review of the published research showed that this is the first study to evaluate the influence of factors on FeNO levels in a group of children with high FeNO values ≥ 20 ppb. Second, standard procedures were performed by uniformly trained medical personnel to measure FeNO in the test children. Third, the study was conducted in the second-largest city in Poland with one of the highest air pollution rates in Poland and Europe [87,88,89]. Fourth, since our study group attended primary schools in Krakow and lived within the city’s administrative boundaries, exposure to air pollution was similar for all participants in the study—the geographic configuration of the basin in which Krakow is located causes accumulation and maintenance of air pollutants in Krakow with a relatively even distribution within the city [90,91]. In addition, all FeNO measurements were made in April over three weeks. Finally, fifth, thanks to the homogeneous age group of the respondents, age does not affect the conclusions obtained.

The drawbacks of this study should also be considered. First, the cross-sectional nature of this study makes it impossible to address the inference of causality. However, the measurement performed at the screening and diagnostic stage of the study, in which all children obtained high FeNO levels, certainly strengthens our results. Second, we used BMI to assess overweight and obesity in children; although it is not the gold standard for assessing body composition in children, BMI categories were determined according to the International Obesity Task Force for Children criteria. Moreover, the majority of studies assessing FeNO level with body weight in both healthy and respiratory children use the BMI index. Third, the survey did not assess the average daily exposure to tobacco smoke among the surveyed children. Fourth, comorbidities in children were reported by parents in the questionnaire, and although the instructions stated that those diseases identified by the doctor should be included, self-reporting of comorbidities by parents may lead to underestimation (in the case of asymptomatic patients) as well as revaluation. Lastly, the residual influence of other immeasurable factors remains possible, although the analyses included several important variables that may influence FeNO levels. Such factors may include regular physical activity or exposure to pollen.

## 5. Conclusions

In conclusion, the level of FeNO in the group of children with FeNO ≥ 20 ppb was significantly higher in children who have asthma, allergic rhinitis, atopic dermatitis, and allergy, and in children taking antihistamines. In multivariate models, it was observed that, regardless of sex, age, BMI, home smoking, or medications, a significantly higher level of FeNO was observed in children suffering from allergic rhinitis or atopic dermatitis or allergy. The strongest relationship was with allergic diseases. However, no significant association was found between FeNO and the incidence of asthma and sex, age, BMI categories, home smoking, and medications. More research is needed to determine if these results can be generalised to other ethnic populations, and future research should consider more factors that may influence FeNO levels.

## Figures and Tables

**Figure 1 medicina-58-00146-f001:**
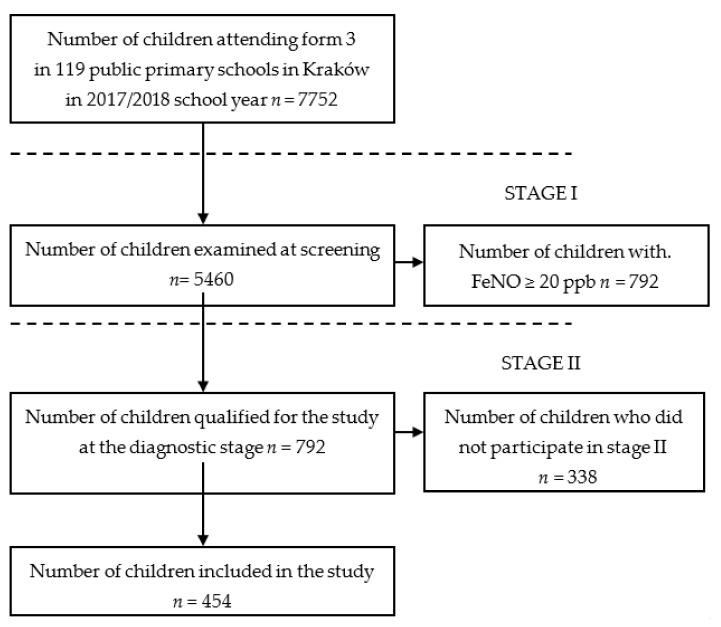
A flow-chart demonstrating the selection of studied groups.

**Table 1 medicina-58-00146-t001:** Baseline characteristics of the study population.

Characteristic	Study Group
Demographic data:	
Age (years) ^a^	8.9 ± 0.46
Girls ^b^	214 (47.14)
Boys ^b^	240 (52.86)
Clinical variables:	
Asthma ^b^	60 (13.21)
Allergic rhinitis (yes) ^b^	144 (31.72)
Atopic dermatitis (yes) ^b^	98 (21.58)
Allergy (yes) ^b^	225 (49.5)
Inhaled glucocorticosteroid (yes) ^b^	30 (6.61)
Antihistamine (yes) ^b^	85 (18.72)
Leukotriene receptor antagonist (yes) ^b^	24 (5.29)
Intranasal corticosteroids (yes) ^b^	38 (8.37)
Presence of cigarette smoke in house ^b^	88 (19.38)
Anthropometric variables:	
Weight (kg) ^a^	33.2 ± 6.89
Height (cm) ^a^	137.7 ± 6.48
BMI-thinness ^b^	35 (7.71)
BMI-normal ^b^	306 (67.4)
BMI-overweight ^b^	88 (19.38)
BMI-obesity ^b^	25 (5.51)
FeNO measurement results:	
FeNO (ppb) (Second stage) ^c^	26 (17–46)

Date is presented as ^a^ mean ± SD; ^b^ n (%); ^c^ median (Q1–Q3). BMI: body mass index; FeNO: fractional exhaled nitric oxide.

**Table 2 medicina-58-00146-t002:** Relationship between the analysed variables and FeNO concentration amongst participants.

Variables	FeNO (ppb)	*p*
	Me (Q1–Q3)	
Age (years)	*r* = −0.032	0.493
Sex:		
Girls	24 (16–42)	0.068
Boys	28 (18–49)	
Asthma:		
Yes	30 (22–52)	0.04
No	25 (16–45)	
Allergic rhinitis:		
Yes	33 (20–55)	<0.001
No	23 (16–41)	
Atopic dermatitis:		
Yes	32 (21–56)	0.002
No	24 (16–44)	
Allergy:		
Yes	33 (21–55)	<0.0001
No	21 (14–34)	
Inhaled glucocorticosteroid:		
Yes	29 (23–42)	0.275
No	26 (17–46)	
Antihistamine:		
Yes	35 (19–53)	0.008
No	24 (17–44)	
Leukotriene receptor antagonist:		
Yes	36 (21–67)	0.092
No	26 (17–45)	
Intranasal corticosteroids:		
Yes	30 (16–50)	0.283
No	25 (17–45)	
Presence of cigarette smoke in the house:		
Yes	29 (19–50)	0.352
No	26 (16–46)	
BMI category:		
thinness	27 (18–38)	0.732
normal	27 (17–47)	
overweight	25 (17–41)	
obesity	22 (14–49)	

Me: median; Q1: lower quartile; Q3: upper quartile; FeNO: fractional exhaled nitric oxide; BMI: body mass index.

**Table 3 medicina-58-00146-t003:** Relationship between FeNO level and factors that may influence its level.

Log FeNO	b(SE)	*p*	b (SE)	*p*	b (SE)	*p*	b (SE)	*p*
Age	−0.076 (0.08)	0.339	−0.077 (0.079)	0.329	−0.061 (0.079)	0.44	−0.06 (0.077)	0.44
Sex (Boys-reference)	−0.1 (0.073)	0.171	−0.086 (0.073)	0.238	−0.122 (0.073)	0.122	−0.05 (0.071)	0.259
Inhaled glucocorticosteroid	−0.047 (0.172)	0.784	−0.008 (0.156)	0.959	−0.013 (0.156)	0.932	−0.071 (0.154)	0.644
Antihistamine	0.134 (0.122)	0.234	0.05 (0.155)	0.666	0.142 (0.11)	0.197	−0.015 (0.122)	0.893
Leukotriene receptor antagonist	0.139 (0.17)	0.413	0.185 (0.168)	0.271	0.154 (0.168)	0.359	0.185 (0.165)	0.262
Intranasal corticosteroids	−0.008 (0.149)	0.959	−0.098 (0.15)	0.514	−0.033 (0.148)	0.822	−0.087 (0.145)	0.548
Presence of cigarette smoke in house	0.132 (0.093)	0.157	0.138 (0.092)	0.135	0.12 (0.092)	0.193	0.141 (0.09)	0.12
BMI category-(thinness reference)								
normal	0.065 (0.138)	0.639	−0.066 (0.137)	0.631	−0.054 (0.136)	0.69	−0.059 (0.134)	0.659
overweight	−0.154 (0.155)	0.32	−0.173 (0.154)	0.26	−0.143 (0.153)	0.35	−0.168 (0.15)	0.263
obesity	−0.16 (0.205)	0.435	−0.15 (0.203)	0.462	−0.138 (0.203)	0.497	−0.178 (0.199)	0.371
Asthma	0.108 (0.124)	0.383	-	-	-	-	-	-
Allergic rhinitis	-	-	0.26 (0.087)	0.003	-	-	-	-
Atopic dermatitis	-	-	-	-	0.266 (0.087)	0.003	-	-
Allergy	-	-	-	-	-	-	0.414 (0.077)	<0.001

b: beta coefficient; SE: standard error.

## Data Availability

The data are available from the corresponding author upon reasonable request.

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
