# Peer review of "Factors Influencing the Concentration of Exhaled Nitric Oxide (FeNO) in School Children Aged 8–9-Years-Old in Krakow, with High FeNO Values ≥ 20 ppb"

_medicina, 2022, doi:10.3390/medicina58020146_

Round 1
Reviewer 1 Report
The manuscript is well presented, describing the significance of the factors influencing FeNO level.
minor Q
- The criteria for selecting children aged 8-9 can be mentioned in the M&M during the screening process.
- Can the authors also comment on the additional seasonal effect on their obtained data, the final diagnosis was carried out during the month of April which is expected that the pollen level surges. Was the baseline FeNO levels (i.e. > 20ppb) changed over the year?
- The authors also should carefully indicate age as a non-influencing factor in their conclusion as the study was only conducted in age group 8-9 yr.
Reviewer 2 Report
The research article is interesting and potentially useful. However, strongly recommend the following additions:
- How can the fractional exhaled nitric oxide test distinguish between chronic respiratory disease recoveries or conditions?
- How long should the children be monitored following the treatment?
- Is there any relationship between children regular physical activity and nitric oxide concentration?
- Table 1 does not contain any details about the boys
